# Emetine in Combination with Chloroquine Induces Oncolytic Potential of HIV-1-Based Lentiviral Particles

**DOI:** 10.3390/cells11182829

**Published:** 2022-09-10

**Authors:** Pavel Spirin, Elena Shyrokova, Valeria Vedernikova, Timofey Lebedev, Vladimir Prassolov

**Affiliations:** 1Department of Cancer Cell Biology, Engelhardt Institute of Molecular Biology, Russian Academy of Sciences, Vavilova 32, 119991 Moscow, Russia; 2Center for Precision Genome Editing and Genetic Technologies for Biomedicine, Engelhardt Institute of Molecular Biology, Russian Academy of Sciences, Vavilova 32, 119991 Moscow, Russia; 3Moscow Institute of Physics and Technology, National Research University, Institutskiy per. 9, 141701 Dolgoprudny, Russia

**Keywords:** emetine, chloroquine, cancer, apoptosis, lentiviral vectors, oncolytic viruses

## Abstract

Chloroquine and Emetine are drugs used to treat human parasitic infections. In addition, it has been shown that these drugs have an antiviral effect. Both drugs were also found to cause a suppressive effect on the growth of cancer cells of different origins. Here, using the replication-deficient HIV-1-based lentiviral vector particles, we evaluated the ability of the combination of these drugs to reduce viral transduction efficiency. We showed that these drugs act synergistically to decrease cancer cell growth when added in combination with medium containing lentiviral particles. We found that the combination of these drugs with lentiviral particles decreases the viability of treated cells. Taken together, we state the oncolytic potential of the medium containing HIV-1-based particles provoked by the combination of Chloroquine and Emetine.

## 1. Introduction

Emetine (EME) is an alkaloid initially isolated from the medicinal plant Psychotria ipecacuanha and has been used in the clinic as an anti-amoebic, emetic, expectorant, and spermicide drug. It has been shown that among all alkaloids extracted from the ipecac roots, EME, being the most present alkaloid in ipecac, inhibits HIV-1 replication at relatively non-toxic or low toxic nanomolar concentrations. According to molecular docking analysis, it was suggested that EME might directly bind to HIV reverse transcriptase (RT), resulting in its inhibition [1]. The antiviral action of EME against different RNA and DNA viruses was shown [2]. In addition, it inhibits the replication of Zika and Ebola viruses [3] and human cytomegalovirus (CMV) [4]. Recently, several studies were performed, and the significant efficiency of EME against the SARS-CoV-2 virus both in vitro and in vivo was evaluated [5,6,7,8]. EME is able to bind irreversibly to the 40S ribosomal subunit, resulting in elongation termination. Thus, it is widely used to study mechanisms of translation [9,10,11].

The antiviral action of the well-known anti-protozoan drug Chloroquine (CQ) and its analogs was studied in vitro and in vivo on different viruses: African swine fever virus [12], HIV-1 [13], SARS-CoV [14,15], Influenza A [16], Chikungunya [17], Ebola [18,19], Zika [20], and, recently, on SARS-CoV-2 [21,22,23]. A mechanism of antiviral action for various viruses is common and based on elevating the pH of acidic intracellular organelles, such as endosomes/lysosomes, essential for virus replication. In the context of the COVID-19 pandemic, the application of CQ and its derivative Hydroxychloroquine (HCQ) was widely discussed to treat SARS-CoV-2 infection [14]. Although it has significant inhibitory efficiency in vitro, it was failed to translate it to in vivo [24]. Because of the strong methodological problems performed in this clinical trial study [25], the possibilities of CQ application to treat COVID-19 remain under discussion [26].

Firstly, the anti-neoplastic activity of CQ was noticed during a mass, long-term anti-malaria trial when the incidents of Burkitt’s lymphoma within CQ treated patients significantly decreased [27]. Burkitt’s lymphoma is known to be associated with Epstein-Barr virus (EBV) infection. Within the last two decades, the anti-neoplastic action of CQ and its analogs was shown for various neoplasms [28,29], including leukemia [30], osteosarcoma [31], prostate cancer [32], glioblastoma [33], and lung cancer [34]. The anti-neoplastic action of CQ and its analogs is mainly based on the inhibition of autophagy. When CQ enters the lysosomes, it becomes protonated because of the low pH within the lysosome. Accumulation of the protonated form of CQ leads to less acidic conditions and disrupted lysosomal function [35,36]. Thus, the antiviral and anti-neoplastic mechanism of CQ action is primarily similar.

Several clinical trials were performed to study the possible therapeutic effects of CQ or HCQ alone or in combination with other anti-cancer agents in various types of cancer [37,38,39,40,41,42]. In addition, the anti-cancer activity of EME was also found in different studies, revealing that various mechanisms may be activated by this drug [43,44].

Interestingly, many molecular mechanisms that are used in intracellular stages of virus replication and involved in the response of the human cells to infection are often the same as those which are deregulated in neoplastic cells in different types of cancer. These include cell cycle regulation mechanisms [45], apoptosis [46,47,48], autophagy [49,50,51], the DNA repair system [52,53,54,55,56,57], cytoskeleton organization mechanisms [58], the ubiquitin–proteasome system (UPS) [59,60,61,62], and immune system response [63,64].

It has been considered that rearrangements in these processes in cancer cells make them preferred for infection by certain types of viruses (herpes simplex virus (HSV), poliovirus, vesicular stomatitis virus, and vaccina virus) and more sensitive to the lytic action of viruses compared to healthy cells. This reflects the general point of oncolytic therapy. Recently, the first oncolytic virus therapy (T-VEC) based on the genetically modified herpes simplex virus, type 1 (HSV-1), was approved by US FDA to treat melanoma [65].

The abovementioned information paves the way to study the oncolytic potential of other known viruses and develop strategies to modulate their cytotoxic action against neoplastic cells.

As it was shown previously, HIV-1 causes pro-apoptotic [48] and cytostatic action on infected cells. The cell cycle deregulation is induced by the HIV-1 or HIV-2 Vpr protein, resulting in G2 arrest of the cell cycle via Cyclin B inhibition [66]. Furthermore, it was shown that the HIV-1 Vpr protein might directly bind to transcription factor Sp1 [67], which is the central regulator of expression of a huge number of genes encoding proteins involved in neoplastic transformation. The interplay between HIV-1 and autophagy pathways is dual and delicate. It utilizes autophagy to enhance viral yields at the early stages while inhibiting autophagy at the late stages [50]. The interplay of HIV-1 with host cell UPS and DNA repair is closely associated to virus replication [54,68].

We suggest that in some conditions, which possibly may be caused by the use of drugs targeting these mechanisms, the HIV-1 or its replication-deficient recombinant forms may cause a suppressive effect on cancer cells.

Using the model system based on the replication-deficient HIV-1 viral particles, we studied the antiviral action of EME and CQ taken alone and in combination. Furthermore, we examined the cytotoxic action of medium containing replication-deficient HIV-1 on neoplastic cells when added in combination with EME and CQ.

## 2. Results

### 2.1. Emetine and Chloroquine Suppress Viability of Cancer Cells

First, we measured the cytotoxic action of CQ and EME on cells of four human cell lines, namely two lung carcinomas A549 and H1299, a leukemia/lymphoma Jurkat cell line, and a human embryonic kidney cell line HEK293T, all widely used in cancer research. The cells were counted after 48 h incubation with CQ and EME. Half-maximal inhibitory concentrations (IC50) were determined for both drugs (Figure 1A). We found that both compounds caused a significant toxic effect on all used cell lines. EME suppressed the viability of cells in nanomolar concentrations. CQ efficiently suppressed viability in micromolar concentrations. The most sensitive both to Emetine and Chloroquine were HEK293T cells, and the less sensitive to the cytotoxic action of EME and CQ were H1299 cells (Figure 1A). For further experiments, we used the drugs in concentrations up to IC50, depending on the cell line.

The transduction efficiency of GFP encoding replication-deficient HIV-1 particles added to the Jurkat, HEK293T, A549, and H1299 cells not pre-treated with CQ and EME is represented in Appendix A.

### 2.2. Combination of Emetine and Chloroquine Suppresses the Infection Efficiency of Replication-Deficient HIV-1 Virus

Next, we evaluated the antiviral action of CQ and EME used in different concentrations. For that, cells were pre-treated with CQ or EME alone or in combination for 24 h (Figure 1B). After that, the medium containing replication-deficient HIV-1 particles (RD HIV-1) bearing the marker gene encoding eGFP and pseudotyped with the G-protein of the vesicular stomatitis virus (VSV-G) was added to achieve not more than 60% of infected cells (Appendix A). At 48 h post infection, the cells were analyzed with flow cytometry. The percentage of infected cells and mean fluorescence intensity (MFI) were detected. The percentage of GFP-positive cells and the MFI in samples treated with drugs were normalized to the percentage of GFP-positive cells in untreated control (represented as 100%) and shown as the relative percentage and relative MFI in the bar plots (Figure 2).

The absolute percentage of transduction efficiency for Jurkat, HEK293, A549, and H1299 cells and an example of the gating scheme is provided in Appendix A.

We found that pre-treatment of cells with EME or CQ alone in concentrations close to IC50 (20 nM for EME and 25 μM for CQ) caused a significant reduction in percentage and MFI of infected Jurkat cells (Figure 2A).

Figure 2 represents the antiviral action of several combinations of drugs when both are taken in low toxic concentrations and the combinations of drugs when one is taken in low or non-toxic concentration and the other is used in relatively high or toxic concentration (higher than IC50), leading to increased antiviral action.

Importantly, we verified that the drugs alone or taken in combination do not affect eGFP fluorescence. For that we obtained the Jurkat cell line stably transduced with vector LeGO G2 encoding GFP. Seven days after, these cells were treated with CQ or EME or their combinations. We found that treatment with drugs did not cause any changes in MFI or percentage of GFP-positive cells (Appendix A). That means that the reduction in the percentage of GFP-positive cells and MFI of the whole population is caused by the suppression of transduction efficiency of viral particles bearing the GFP-encoding gene, but not the GFP biogenesis, which may be possibly affected by the drugs.

We found that the pre-treatment of cells with a combination of EME with CQ followed by virus addition causes an approximate three-fold decrease in the percentage of GFP-positive cells in the whole population of cells treated with drugs, and decreases its MFI compared to cells not pre-treated with CQ and EME but infected with the virus. Importantly, CQ in a non-toxic concentration of 5 μM does not affect infection efficiency when taken alone, but in combination with 20 nM of EME causes the same three-fold reduction in fluorescence intensity (Figure 2A,B). The decrease in fluorescence intensity and percentage of cells infected with RD HIV-1 was also detected when HEK293T (Figure 2C,D), A549 (Figure 2E,F), and H1299 cells (Figure 2G,H) were treated with EME or CQ alone or in combinations. The most pronounced suppression of infection efficiency was observed when the relatively high concentrations of drugs (close to IC50) were used. Interestingly, treatment with EME alone in concentrations up to 20 nM and CQ alone in concentrations up to 10 μM did not cause significant suppression in the percentage of HEK293T cells infected/transduced with RD HIV-1. When EME 20 nM and CQ 10 μM were taken in combination, we observed about a 40% decrease in the percentage of GFP-positive cells (Figure 2C). Notably, this combination caused a significant (up to 95%) reduction in MFI, suggesting that treatment with CQ and EME reduced the virus infectivity. Even the combination of CQ in a lower concentration of 2.5 μM with EME 20 nM significantly decreased the MFI of cells infected with RD HIV-1.

CQ added alone in the non-toxic concentration of 5 μM to lung carcinoma A549 or H1299 cells caused pronounced suppression of infection efficiency (Figure 2E,G). EME added alone in a low toxic concentration of 10 nM to A549 cells did not affect infection efficiency, even taken in combination with CQ 5 μM. A slight enhancement of the antiviral effect of CQ (10 μM; close to IC50) was detected when EME (10 nM) was used. EME added in higher concentrations (40 nM) efficiently suppressed the infection, and an antiviral effect was observed when EME was used in combination with CQ 5 μM (Figure 2E,F). The main restriction of this experiment is that the 40 nM concentration of EME is much higher than IC50 and is mostly toxic for A549 cells.

On the contrary, we found that EME taken in low toxic concentrations up to 40 nM significantly enhanced the antiviral action of CQ used in the non-toxic concentration of 5 μM (Figure 2G).

Taken together, we found that different combinations of EME and CQ may be sufficient to suppress the infection caused by replication-deficient HIV-1 particles. The most pronounced antiviral action was observed when drugs were used in relatively high concentrations close to IC50. The increase in drug concentrations resulted in higher toxic effects when used in combination with the lentiviral particles. We suggest that the decrease in the percentage of infected GFP-positive cells may be primarily associated with cell death induced by the medium containing HIV-1 particles when combined with EME and CQ.

### 2.3. Emetine in Combination with Chloroquine Affects Viability of Cancer Cells Transduced with RD HIV-1

To evaluate the effect of RD HIV-1 treated with drugs on the viability of cancer cells, we performed a 24 h pre-treatment of cells with EME and CQ. Four cell lines, Jurkat, HEK293T, A549, and H1299, were treated with both drugs in different concentrations and drugs added alone followed by the addition of RD HIV-1 to achieve not more than 60% of transduced cells in the population, as it was performed in previous Section 2.2. As a control, we used the cells pre-treated with drugs followed by conditioned DMEM addition instead of RD HIV-1. The cells were counted and dose–response matrixes were obtained (Figure 3A and Figure 4A). We found that both drugs, EME and CQ, significantly reduced cell growth when added alone. After that, the synergy distribution plots were generated and analyzed with the SynergyFinder v2.0 web application. Using the ZIP model to capture the drug interaction relationships, the most synergistic areas were detected, and the synergy scores were calculated (Figure 3B,E and Figure 4B,E). The synergy scores up to 10 mean that the interaction between two drugs is likely to be additive. Scores larger than 10 mean that the interaction between two drugs is likely to be synergistic.

We found that treatment with a combination of EME, CQ, and RD HIV-1 causes a pronounced synergistic suppressive effect on the growth of human leukemia Jurkat cells (Figure 3B). Importantly, when taken alone, the RD HIV-1 did not affect the cell survival even when the high volumes of viral stocks were taken (Appendix A).

HEK293T cells, which are widely used to study neoplastic transformations, were also synergistically suppressed by the combination of these drugs in the presence of the RD HIV-1 (Figure 3D). Importantly, we found that non-toxic concentrations of EME (5 nM) combined with CQ in the low toxic concentration of 7.5 μM significantly reduced the Jurkat cell growth when RD HIV-1 was added compared to the control sample without a virus (Figure 3C upper panel). The combinations of drugs in higher concentrations were also more cytotoxic when RD HIV-1 was added (Figure 3C bottom panels). A similar synergistic suppressive effect was shown when EME in a non-toxic concentration (10 nM) and CQ taken in a low toxic concentration (2.5 μM) were added to HEK293 cells in combination with RD HIV-1 (Figure 3F).

Both drugs, EME and CQ, were shown to significantly suppress cell growth when added alone (Figure 4A,C). The joint additive action of EME and CQ in combination with RD HIV-1 was observed when lung carcinoma cells A549 and H1299 were treated (Figure 4B,E). Importantly, when we added the drugs to the cells 24 h post infection, the sensitivity of cells to EME and CQ was not changed compared to cells not transduced with lentiviral particles. That possibly means that integrated provirus encoding GFP do not cause any additional effect on survival of cells treated with drugs. These results suggest that the cytotoxic action of viral stocks in combination with drugs is implemented on the first stages of viral life cycle prior to integration into the host cell genome.

Although no synergistic action of EME and CQ used in different combinations with RD HIV-1 was detected on A549 cells (Figure 4C), we found that treatment of H1299 p53-deficient cells with a non-toxic concentration of EME 10 nM and CQ 5 μM caused significant synergistic reduction in cell growth 48 h post addition of RD HIV-1 (Figure 4F upper panel). The higher concentrations of CQ and EME used in combinations were also shown to synergistically suppress H1299 cell growth followed by virus addition compared to control samples not treated with a virus (Figure 4F bottom panel).

### 2.4. Emetine and Chloroquine Induce Apoptosis of Cells Treated with RD HIV-1

Next, a double staining assay (Annexin V/PI) was conducted to determine the apoptotic effect of EME and CQ in combination with RD HIV-1 on cells. For that, Jurkat, HEK293T, A549, and H1299 cells were treated with the combination of drugs in different concentrations followed by infection with RD HIV-1 to achieve not more than ~50% of infected cells in the whole population. We found that treatment of cells with a combination of EME and CQ induce apoptosis (Figure 5A–H). We did not find any significant impact of the virus on the pro-apoptotic action of EME and CQ on Jurkat, A549, and H1299 cells in the whole population. A significant action of RD HIV-1 on apoptosis activation in HEK293T cells treated with CQ and EME was evaluated (Figure 5G).

The percentage of infected cells (GFP-positive) in the whole population treated with CQ and EME followed by RD HIV-1 addition was 52.2% for Jurkat, 53.3% for A549, 54.4% for HEK293T, and 52.9% for H1299. 

Next, we compared the percentage of apoptotic cells in the whole population of cells treated with RD HIV-1, CQ, and EME with a sub-population of cells that were infected (GFP-positive) and with a sub-population of cells which were not infected (GFP-negative). We found that the percentage of apoptotic cells in the sub-population of infected Jurkat, A549, and HEK293T cells was significantly higher than the percentage of apoptotic cells in the non-infected sub-population. The rate of apoptotic cells in the whole population treated with EME, CQ, and RD HIV-1 was significantly lower than in the sub-population of cells that were indeed infected with RD HIV-1 (GFP-positive) (Figure 5J–P). Interestingly, we did not determine any impact of RD HIV-1 on apoptosis activation of H1299 p53-deficient cells treated with EME and CQ (Figure 5Q).

## 3. Discussion

Here, we found that treatment of cells with EME and CQ significantly reduces the infection efficiency of replication-deficient HIV-1 (RD HIV-1). We noticed that cells treated with CQ, EME, and RD HIV-1 look less viable when compared to cells treated only with CQ and EME. Notably, the addition of RD HIV-1 alone in concentrations used in this study does not affect the viability of cells. We suggested that the antiviral action of EME and CQ is strongly associated with its cytotoxic action. Trying to use lower concentrations of the drugs, we found that cytotoxic activity of EME and CQ is paradoxically attenuated with RD HIV-1 addition. Furthermore, we found that treatment with a combination of these drugs induces apoptosis of cells infected with RD HIV-1, unlike the cells treated with EME and CQ but treated with RD HIV-1.

These results reveal that RD HIV-1 in the presence of EME and CQ may exhibit oncolytic properties. The mechanisms of oncolytic action of different viruses are not clearly understood. The oncolytic action of FDA-approved TVEC is mainly based on exploiting the disrupted antiviral pathways, which are also involved in neoplastic transformation, primarily the protein kinase R (PKR) and type I interferon (IFN) pathways [69]. The action of other oncolytic viruses is actively studied. The interplay between autophagy in cancer cells and its pivotal role for viruses may partially explain the oncolytic potential of various viruses [70]. The involvement of the ubiquitin–proteasome system in cancer cell viability and virus replication was utilized in a study demonstrating the use of proteasomal inhibitor bortezomib in combination with oncolytic HSV-1, which synergistically inhibited the growth of intracranial tumors of glioma and head and neck cancer in vivo [71]. In drug-resistant tumor models, it was shown that the oncolytic potential of VSV can be reversibly stimulated by combination with epigenetic modulators, such as the histone deacetylase inhibitor vorinostat [72]. These partially explain the possible mechanisms, which may be associated with the oncolytic action of various viruses.

The pro-oncolytic action of replication-deficient HIV-1 induced by EME and CQ may be associated with antiviral and anti-cancer mechanisms of these drugs. The antiviral action of EME and CQ combination was shown for the first time but mostly it was expected, unlike the oncolytic activity of RD HIV-1 induced by EME and CQ which was not apparent. The provided results may be partially explained by the substantial experimental data obtained by other groups. Both drugs used in this study, EME and CQ, were early shown to cause pronounced inhibitory action on replication of different viruses, including HIV-1. The antiviral and tumor-suppressive action of CQ is considered to be associated with autophagy inhibition in response to pH elevation caused by CQ [49]. The anti-cancer effect of CQ is generally explained via autophagy deregulation [35]. Notably, the more specific mechanisms targeted by CQ in cancer cells were defined earlier. As an example, CQ was shown to cause inhibitory action on AKT and ERK signaling. Due to this, CQ was also shown to effectively overcome the innate resistance of wild-type EGFR non-small-cell lung cancer cells to the EGFR inhibitor erlotinib [73]. CQ was shown to induce p21 expression alone and in combination with cisplatin to overcome the chemoresistance in ovarian cancer cells [74]. CQ induces cell death by activating the p53 pathway in glioblastoma [75]. The antiviral action of EME against HIV-1 was shown to be caused by inhibition of HIV-1 reverse transcriptase [1]. It has been reported that Emetine inhibits translation of viral proteins by blocking the 40S ribosomal protein S14 in host cells [4,9,76]. The mechanism of its anti-protozoal action is still under discussion. One of the possible effects is based on its possibility to intercalate into trypanosome DNA, resulting in their death [77].

The anti-cancer action of EME is of particular interest. EME was shown to induce caspase-3, caspase-9/6, and caspase-8, followed by apoptosis induction in different human cancer cell lines [78]. Furthermore, the antitumor activity of EME was shown to be associated with its inhibitory effects on the Wnt signaling pathway [43]. The inactivating mutations in the Von Hippel-Lindau (VHL) tumor suppressor factor led to stabilization of HIF-1ɑ and HIF-2ɑ factors, which is often associated with the development of renal cell carcinomas. EME was shown to promote VHL-independent down-regulation of HIF-2ɑ in several renal cancer cell lines [44]. Recently, EME was identified as a therapeutic agent for pulmonary arterial hypertension (PAH). Its anti-PAH action was also connected with reduction in HIF-1ɑ and HIF-2ɑ protein levels and their downstream targets PDK1 (pyruvate dehydrogenase kinase 1), RhoA (Ras homolog gene family, member A), ROCK1 and ROCK2 (rho-associated coiled-coil containing protein kinases 1 and 2), and their downstream CyPA (cyclophilin A) and Bsg (basigin) [79]. As it was previously mentioned, EME is widely used to study translation, because of elongation termination via inhibiting the 40S ribosomal subunit [11]. Importantly, in translation studies relatively high concentrations of EME (1–300 μM) are used to repress the elongation. These concentrations are much higher than toxic concentrations of this drug used in our study and the studies conducted by other groups. This suggests the other targets are responsible for the cytotoxic and tumor-suppressive activity of Emetine.

Summarizing the data of our experimental results indicates that EME is necessary for stimulating the cytotoxic action of RD HIV-1 on neoplastic cells, and previously published data reveal that EME inhibits HIV-1 RT [1]; therefore, we suggest that the oncolytic action of RD HIV-1 induced by a combination of EME and CQ becomes implemented on the initial stages of virus replication, foregoing provirus synthesis. These may be the steps associated with virus entry or endosomal trafficking. Interestingly, the autophagy suppression induced by EME was earlier described and predicted in several studies [3,80,81]. Inhibition of autophagy was shown to restore and enhance the sensitivity of various cancer cells to anti-cancer treatment [82]. Autophagy plays an important role in the early steps of the HIV-1 life cycle associated with endosomal trafficking. Autophagy is a pivotal mechanism associated with the survival of cancer cells. The combination of EME and CQ induces its deficiency in cancer cells. We speculate that exploitation of autophagy in cancer cells after treatment with medium containing replication-deficient HIV-1 particles leads to increased autophagy deficiency in treated cancer cells, resulting in enhancement of their cell death. Importantly, the described action may be caused not only by a virus, and the input of other factors in viral stocks cannot be excluded, thus it needs further investigation.

Here, we state the pro-oncolytic activity of the viral stocks containing HIV-1 replication-deficient viral particles, which may be potentially induced by a combination of the FDA-approved drugs Emetine and Chloroquine. This may be of significant interest in the context of drug repurposing strategies for the development of novel anti-HIV-1 and cancer therapy approaches.

## 4. Materials and Methods

### 4.1. Cell Lines, Culture Conditions, and Reagents

Embryonic kidney cells HEK293T and human lung cancer cells A549 and H1299 were cultured in a DMEM growth medium. The human leukemia cancer cells Jurkat were cultured in RPMI-1640 growth medium. Cells were cultured in a humidified atmosphere with 5% CO2 at 37 °C, and all growth media were also supplemented with 2 mM L-glutamine, 100 units/mL penicillin, 100 µg/mL streptomycin, and 1 mM sodium pyruvate and 10% fetal bovine serum (FBS). DMEM, RPMI-1640, FBS, penicillin/streptomycin, sodium pyruvate, and L-glutamine were purchased from Gibco (ThermoFisher Scientific, Waltham, MA, USA). The cells were continuously kept in culture for a maximum of 2 weeks after thaw. For apoptosis assay, cell viability assay and fluorescence detection using flow cytometry of adherent cell lines (HEK293T, A549, and H1299) were prepared for analysis by washing with ×1 PBS solution (Merck KGaA, Darmstadt, Germany) and treatment with ×1 Trypsin-EDTA solution (Merck, KGaA, Darmstadt, Germany). Jurkat and HEK293T cell lines were a gift from Prof. Dr. Boris Fehse, UKE UNI, Hamburg, Germany. A549 and H1299 cell lines were gifted by Prof. Dr. Thomas Dobner, Heinrich-Pette Institute at the Leibniz Institute for Experimental Virology. Drugs used in the current study were Chloroquine (C6628, Sigma Aldrich, Burlington, MA, USA) and Emetine (E2375, Sigma Aldrich, Burlington, MA, USA).

### 4.2. Lentiviral Particle Production

The stocks containing VSV-G pseudotyped lentiviral particles were obtained via calcium phosphate transfection by co-transfection of HEK293T packaging cells with 10 µg LeGO-C2 plasmid and packaging plasmids (10 µg of pMDLg/pRRE, 5 µg of pRSV-Rev, and 2 µg VSV-G). At 8 h after transfection, the medium was changed with DMEM containing 20 mM HEPES (ThermoFisher Scientific, Waltham, MA, USA). After 12 h, the supernatants containing VSVG pseudotyped viral particles were collected, filtered through a 0.22 µm filter (Millipore), and stored at −80 °C. The supernatant containing pseudotyped vector particles was titrated using HEK293T cells as described before [83]. Titer was 2.6–2.9 × 10^6^ units/mL.

### 4.3. Analysis of Fluorescence

Cells were transduced with lentiviral particles to achieve a ~ 50% transduction rate. At 48 h after lentiviral particles were added, the transduction rates were verified by flow cytometry using LSRFortessa flow cytometer (BD Biosciences, San Jose, CA, USA) and analyzed with FlowJo v10.0.7 software (FlowJo LLC, Ashland, OR, USA). All images were obtained by EVOS FL Cell Imaging System (ThermoFisher Scientific, Waltham, MA, USA) using ×10 magnification lenses.

### 4.4. Cell Viability Assay

The Jurkat, HEK293T, and H1299 cells (5000 cells/well in 200 µL/well) and A549 cells (10,000 cells/well in 200 µL/well) were plated into 48-well plates and treated with 0–25 µM of Emetine (EME) and 0–50 µM of Chloroquine. Chloroquine was dissolved in PBS, and Emetine was dissolved in H_2_O. Cellular viability was assessed 48 h and 72 h post drug treatment using 0.4% Trypan blue solution (ThermoFisher Scientific, Waltham, MA, USA) with exclusion in a 1:1 ratio in the Neubauer chamber. IC50s were calculated by nonlinear regression with variable slope (four parameters) and robust fitting using The GraphPad Prism software v.9.3.1 (GraphPad Software, San Diego, CA, USA).

### 4.5. Drug Combination Analysis

The effects (synergistic versus antagonistic) of Emetine with Chloroquine were determined using web-based SynergyFinder 2.0 software (https://synergyfinder.fimm.fi) (accessed on 9 September 2022) [84]. Cells were treated with Emetine (0–20 nM for Jurkat and HEK293T, 0–40 nM for A549 and H1299) and Chloroquine (0–10 µM for HEK293T, 0–25 µM for A549 and H1299, and 0–30 µM for Jurkat) alone or in combination and after 24 h cells were infected. Cells were incubated for 72 h, and the viability was measured as described above. The zero interaction potency (ZIP) [85] model was used for the reference.

### 4.6. Analysis of Apoptosis

The cells were plated into 48-well plates in concentrations of 5000 cells per well for Jurkat, HEK293T, and H1299 and 10,000 cells per well for A549 in the final volume of 150 µL and treated with 0–10 nM of Emetine and 0–10 µM of Chloroquine. At 24 h, cells were transduced with lentiviral particles to a final volume of 250 µL to achieve ~ 50% transduction rate. Apoptosis was measured by double staining with Annexin V-FITC (Molecular 429 Probes) [86] and propidium iodide (PI) [87] 48 h after transduction. All measurements were performed on an LSR Fortessa flow cytometer (BD Biosciences, San Jose, CA, USA). Analysis of apoptosis rate was performed with FlowJo software 10.0.7 (FlowJo LLC, Ashland, OR, USA).

### 4.7. Data and Statistical Analysis

The GraphPad Prism software v.9.3.1 (GraphPad Software, San Diego, CA, USA) was used for statistical analysis. The unpaired Student’s *t*-test (for comparison of two groups) or one-way ANOVA was used to compare the treated groups with a control group. All experiments were performed in triplicate. Statistically significant differences (*) were assumed if *p* < 0.05.

## Figures and Tables

**Figure 1 cells-11-02829-f001:**
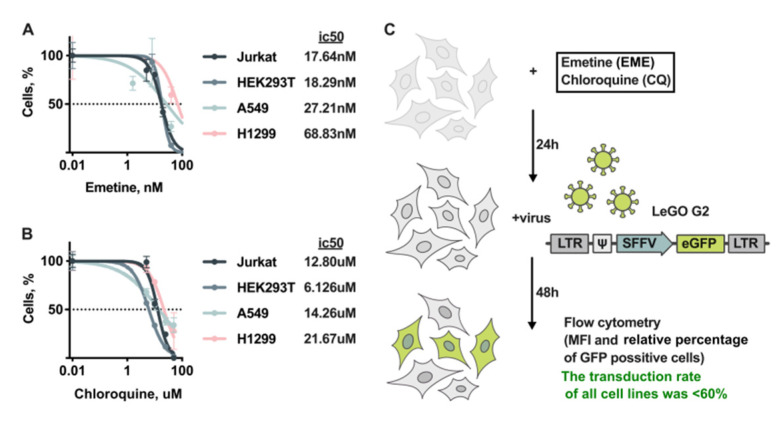
Cytotoxicity of EME and CQ. Scheme of the experiment. Viability of human cancer cells treated with (**A**) 0–25 nM of EME and (**B**) 0–50 µM of CQ; dotted line shows the level where the percentage of alive cells reaches 50%; cell viability was measured 48 h post drug treatment using Trypan blue exclusion. (**C**) Scheme of the experiment: cells were pretreated with EME and CQ for 24 h and transduced with lentiviral particles pseudotyped with G-protein of vesicular stomatitis virus (VSVG). The recombinant plasmid LeGO G2 was used (LTR—long terminal repeats, ψ—packaging signal, SFFV—spleen focus-forming virus promoter, eGFP—marker gene encoding green). Percentage and fluorescence intensity of eGFP-positive cells were measured by flow cytometry 48 h after transduction (see next section).

**Figure 2 cells-11-02829-f002:**
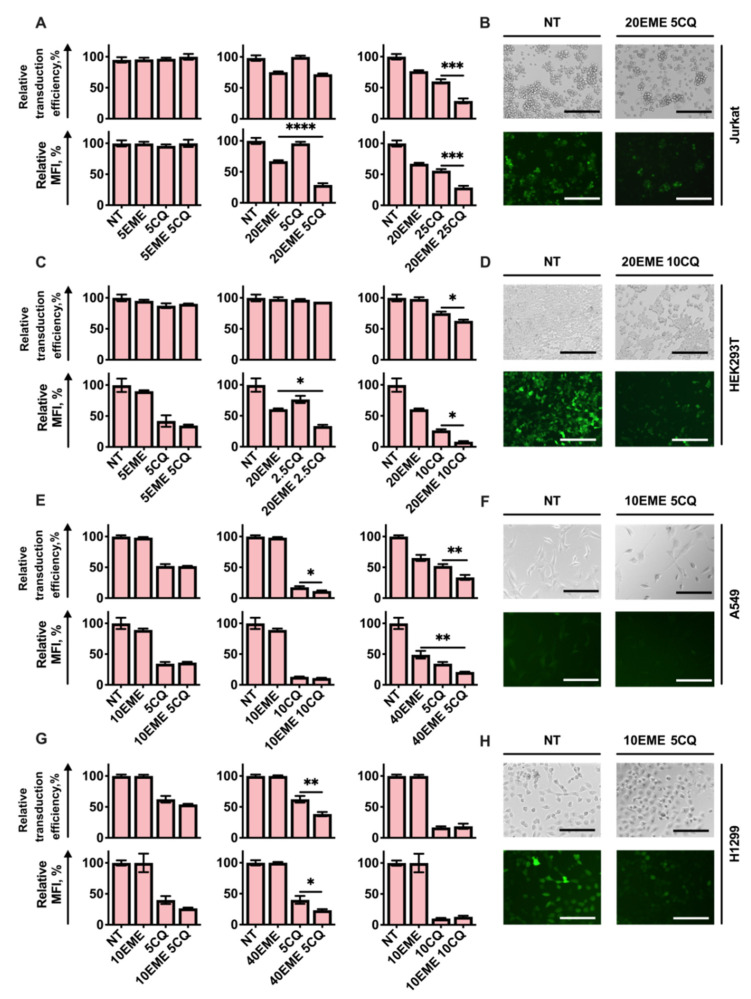
The quantitative differences in antiviral action between Emetine and Chloroquine and its combinations. Jurkat and HEK293T cell lines were pre-treated with EME in combination with CQ and 24 h after that were infected with RD HIV-1. (**A**,**C**,**E**,**G**) Graphs represent the relative percentage of transduced cells and relative MFI level in cells pre-treated with EME and CQ compared to non-treated cells. (**B**,**D**,**F**,**H**) Phase-contrast and fluorescence images of cells treated with EME and CQ compared to non-treated cells. Scale bars 200 μm. Asterisks: * *p* < 0.05; ** *p* < 0.01; *** *p* < 0.001; **** *p* < 0.0001.

**Figure 3 cells-11-02829-f003:**
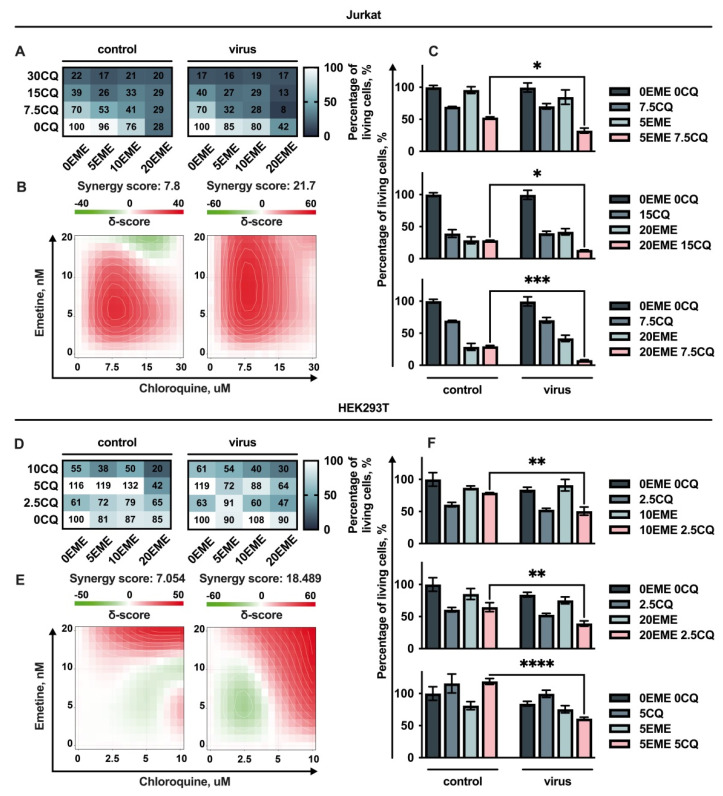
Analysis of the effect of Emetine in combination with Chloroquine. Jurkat and HEK293T cell lines were co-pre-treated with EME in combination with CQ and infected 24 h after. Viability was measured 72 h post drug treatment using Trypan blue exclusion; (**A**,**D**) dose–response matrix represents the percentage of viable cells compared to non-treated control cells in both control (not-infected) and virus (infected) groups. (**B**,**E**) Synergy plots represent the effect of the drug combination that was calculated and visualized using SynergyFinder v2.0 software and a ZIP reference model. Red regions indicate synergism, white—additive effect, green—antagonism; (**C**,**F**) the graphs represent the viability of cells in % of control for effective combinations. Asterisks: * *p* < 0.05; ** *p* < 0.01; *** *p* < 0.001; **** *p* < 0.0001.

**Figure 4 cells-11-02829-f004:**
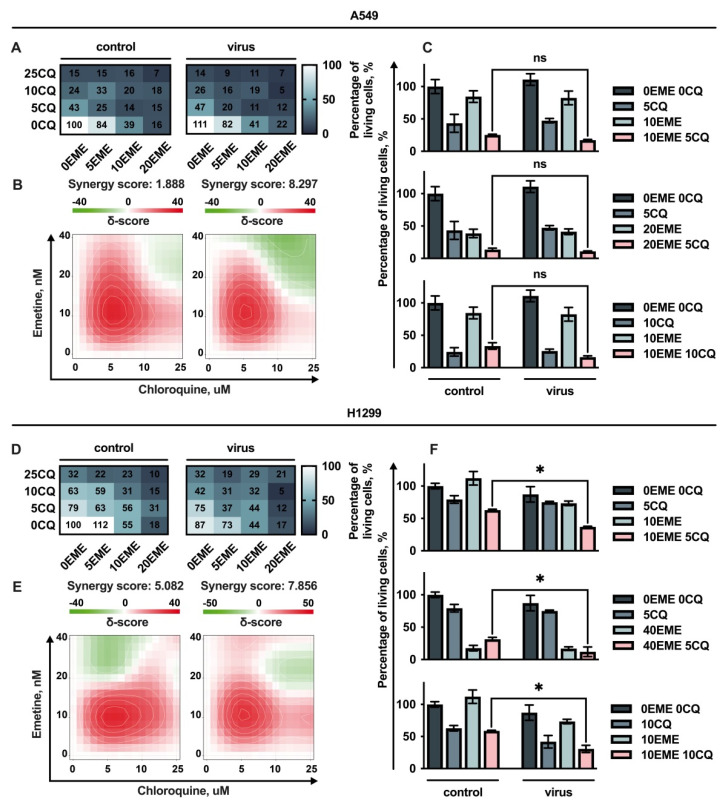
Analysis of the effect of Emetine in combination with Chloroquine. A549 and H1299 cell lines were co-pre-treated with EME in combination with CQ and infected 24 h after. Viability was measured 72 h post drug treatment; (**A**,**D**) dose–response matrix represents the percentage of viable cells compared to non-treated cells in both control (not- infected) and virus (infected) groups. (**B**,**E**) Synergy plots represent the effect of the drug combination (synergism/additive effect/antagonism) that was calculated and visualized using SynergyFinder v2.0 software and a ZIP reference model. Red regions indicate synergism, white—additive effect, green—antagonism; (**C**,**F**) the graphs represent the viability of cells in % of control for effective combinations. Asterisks: ns *p* > 0.05; * *p* < 0.05.

**Figure 5 cells-11-02829-f005:**
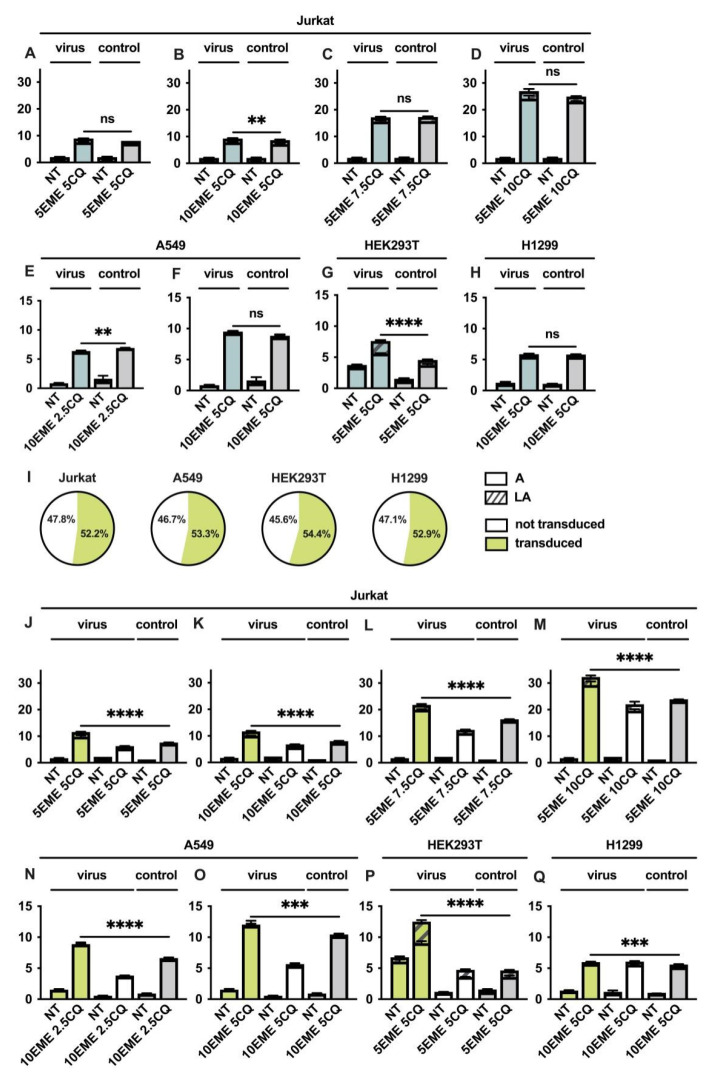
Emetine and Chloroquine induce apoptosis of infected cells. Jurkat, A549, HEK293T, and H1299 cell lines were co-pre-treated with EME in combination with CQ and infected with RD HIV-1 24 h after. (**A**–**H**) The graphs represent a distribution of Annexin V/PI-stained cells 72 h post infection compared to control cells. (**I**) Pie charts represent the percentage of transduced and non-transduced cells 72 h after infection. (**J**–**Q**) The graphs represent a distribution of Annexin V/PI-stained cells 72 h post-infection in transduced and non-transduced cells compared to control cells. A—early apoptosis (Annexin V single-positive cells), LA—late apoptosis (Annexin V/PI double-positive cells). Asterisks: ns *p* > 0.05; ** *p* < 0.01; *** *p* < 0.001; **** *p* < 0.0001.

## Data Availability

Not applicable.

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
