# Peer review of "Emetine in Combination with Chloroquine Induces Oncolytic Potential of HIV-1-Based Lentiviral Particles"

_cells, 2022, doi:10.3390/cells11182829_

Round 1
Reviewer 1 Report (Previous Reviewer 2)
The manuscript “Emetine synergistically with Chloroquine induces oncolytic potential of HIV-1 based lentiviral particles.” is a resubmission of the previously presented version that has taken into account a number of points raised in the first round of comments.
The term HIV contains the word “virus” and HIV virus should not be used. HIV would suffice.
Please modify the EM to EME in some instances such as in line 65 or 100.
You have nicely shown the cytotoxicity of individual drugs in figure 1. How about combined cytotoxicity? Is there any synergistic cytotoxicity in this case?
The figure 1 legend states the EME concentrations in a wrong unit.
In Figure 2 the combination treatments are presented with the transduced cell% and mean fluorescent% to untreated cells. There are simple spreadsheets to show the drug synergy/antagonism such as http://dx.doi.org/10.1016/j.csbj.2015.09.001 or doi: 10.1093/bioinformatics/btx162 and are easy to follow. I suggest using one of these datasheets to present your data.
You have also used the reporter gene to detect the virus infectivity, but the internal control is missing on the effect of the drugs alone or in combination on translation efficiency of a GFP expressing vector up on transfection. This could be used to normalize the values to the infected cells with the non-replicating viruses. In other words, the reduction in fluorescence could be due to mechanisms unrelated to antiviral activities of the drugs in combination.
In the lines 151-155 you have mentioned that using doses of IC50 when drugs were combined reduced the infected cell percentage only 40%. Based on IC50 definition the half of the cells should be dead at these concentrations. How is this possible?
In line 155 to 157 you have suggested that the combination of the drugs reduces the MOI for the virus. This might not be true based on virological definition of MOI. It might be interpreted as increasing of the MOI but still is not a proper expression of the antiviral concept. The MOI is something you define based on the number of particles used to infect a cell. “The infectivity and/or virus replication is blocked by the use of drugs” might be a proper expression since you have not also formally studied the difference between the stages of the virus replication including entry, replication, or infection in general.
In Figure 3 you have nicely shown the effect of the drugs individually and in combinations on the cell faith as well as the effects in the presence of the virus. The experiment showing the effect of only virus is missing from the set to normalize the values to the values obtained from the virus only tests. In this way the efficacy of the virus infection is not known. Since the drugs also have antiviral effect this can cause failure in full CPE formation by the virus.
In fact, the section in lines 224-229 points out to the direction that the presence of the drugs after the infection did not reduce cell survival rate therefore argues against a synergistic effect between virus and drugs. Additionally, it proposes that the provirus genome integration might have caused the resistance to cell killing when drugs were added after the infection which might point out the CPE being related to the early events in the transduction.
You have also talked about the additive effect and synergistic effects which might not be justified by the data you present. They should be discussed to make it clear why the effect is additive or synergistic or both.
Author Response
We are very grateful to the referee for a thorough analysis of our manuscript and substantial questions and suggestions. We have made all the necessary corrections in the text of the manuscript. We also provide answers to questions posed by the reviewer in the *.pdf file attached.

Reviewer 2 Report (Previous Reviewer 1)
The investigation conducted by the authors contributes novel and relevant insights regarding the effects of the antiparasitic drugs Chloroquine and Emetine on the growth of cancer cells. Through experiments, the authors depict the synergistic oncolytic potential of the combination of Chloroquine, Emetine, and HIV-1 based particles.
The authors have made significant changes to the manuscript in response to multiple reviewer recommendations. Hence, the article is now, in my opinion, a well-written article that could be beneficial for future research investigations on cancer growth suppression utilizing anti-viral and antiparasitic drugs and hence is worthy of consideration for publication in the journal cells.
Author Response
We are very grateful to the reviewer for the warm comments and appreciating of our work.
Round 2
Reviewer 1 Report (Previous Reviewer 2)
I shall thank the authors for the revision and the comments.
There are differences in the opinions and I would like to still argue a few points I feel need to be improved.
At a second glance (line 110) it seems that the issue of the cell killing is interpreted as “EME caused suppression of cell growth”. It is not what you have actually studied here since the cell cycle have not been studied in here. I suggest rephrasing this part to avoid misinterpretation.
In the first section of the results as explained in the Figure 1 in the legend in the lines 126-128 “Percentage and fluorescence intensity of GFP positive cells were measured by flow cytometry 48 hours after transduction.” the results have not been presented and this part is missing in the main text of the section 2.1.
It is still not clear what the transduction rate was. you targeted to have not more than 50% of the treated cells (with CQ and EME) being infected as mentioned in the lines 134-136 “After that, the medium containing replication-deficient HIV-1 particles (RD HIV-1) bearing marker gene encoding eGFP and pseudotyped with G-protein of vesicular stomatitis virus (VSV-G) was added to achieve not more than 50% of infected cells.”. How do you explain the 100% transduction efficiency shown in Figure 2A, C, E, and G?
Now you argue that it is a relative number to untreated cells, then it is normalized to the number of untreated cells. This cannot be true also since the untreated cells do not show any cytotoxicity due to drug treatments, therefore you would expect a huge difference between treated cells and non-treated cells only based on drug cytotoxicity (as you suggested in figure 1, the IC50 for CQ is 12.8 for example for Jurkat cells and 17.64 for EME). The amounts used for Jurkat cells for example (25CQ and 20EME) would kill more than 50% of the cells in the assay and the total number of transfected cells would decrease significantly. Therefore, it seems that the normalization should be done to double drug treated cells not the un-treated cells or live cells at the best in the same treatment. Otherwise, this will generate significant results only because of the cytotoxicity alone. This is what I am trying to argue (in the last round of comments also) and still think that the normalization to un-treated cells is not the right way to present the data. Some of the synergy calculating programs take into account the cytotoxicity in combination when calculating the effect and can be used to show the data. You can easily visualize this in the pictures in Figure 2B. Imagine you show the number of live cells in the drug treated cells.
It seems that you have selectively shown different treatments in the Figure 2B, D, F, and H than the highest concentrations used in the assay. Could you please also justify why 20EME 10CQ (not the highest) was shown for Jurkat cells but 10EME 5CQ was shown for A549 cells (the lowest concentrations)?
When you say (in line number153) “The combination of these drugs causes about three-fold decreases in percentage and MFI of infected cells.” do you mean the expression of the GFP is reduced in infected cells compared to GFP expression in drug-treated cells? In other words, the level of expression is reduced regardless of the absolute number of infected cells. I think you mean the MFI is three folds reduced compared to un-infected cells. If not, then this is the next level of the effect since the expression of GFP is also reduced in the presence of the drugs, which points out to the downstream events in the gene expression when cells are transduced with the GFP coding plasmid LeGO G2.
Language: Please perform some editing to the text. Examples of which are below:
Line 107-109) The cells were counted after 48h exposition with CQ or EME, and half-maximal inhibitory concentrations (IC50) were determined for both drugs (Figure 1A).
Author Response
Reviewers comment.I shall thank the authors for the revision and the comments.
There are differences in the opinions and I would like to still argue a few points I feel need to be improved.
Answer. Thank you very much, we are absolutely agree with the comments. The misunderstanding of some points in our manuscript was caused by the insufficient explanation of some parts of the results included. We tried to improve the manuscript and provided the explanations, which, we hope, would be accepted by the reviewer.
Point 1. At a second glance (line 110) it seems that the issue of the cell killing is interpreted as “EME caused suppression of cell growth”. It is not what you have actually studied here since the cell cycle have not been studied in here. I suggest rephrasing this part to avoid misinterpretation.
Answer to point 1. Thank you for this point, we used this term at the beginning of our study we used term «cell growth» to avoid the terms proliferation or cell death, because it was not clear what realy cause decrease in cell quantity. As recommended we rephrased this sentence and excluded term «cell growth» changing it for «cell viability» which is more appropriate in current context.
Point 2. In the first section of the results as explained in the Figure 1 in the legend in the lines 126-128 “Percentage and fluorescence intensity of GFP positive cells were measured by flow cytometry 48 hours after transduction.” the results have not been presented and this part is missing in the main text of the section 2.1.
Answer to point 2. On the Figure 1 C we provided the scheme of the experiment to demonstrate the steps and time line of the study. We agree with the reviewer and now we realized that the lack of the data demonstrating the efficiency of cell line transduction caused the difficulties with interpretation of obtained results at the next steps of our study. We clarified that the percentage of the transduced cells was not more than 50-60% to avoid overinfection at this section of the results, corrected the scheme of the experiments and provided the Supplementary Figure S1 to demonstrate the exact percentage and
MFI of the transduced cells.
Point 3. It is still not clear what the transduction rate was. you targeted to have not more than 50% of the treated cells (with CQ and EME) being infected as mentioned in the lines 134-136 “After that, the medium containing replication-deficient HIV-1 particles (RD HIV-1) bearing marker gene encoding eGFP and pseudotyped with G-protein of vesicular stomatitis virus (VSV-G) was added to achieve not more than 50% of infected cells.”. How do you explain the 100% transduction efficiency shown in Figure 2A, C, E, and G?
Answer to point 3. We provided the dot plots with the percentage of cells transduced with viral particles encoding GFP in Supplementary Figure S1. We agree and understand that this misinterpritation was caused by the absence of clear explanation what the transduction rate was at the first section of our study. We added this information into the text of our manuscript and provided the supplementary figure with this data. We used the different amounts of virus to achieve not more than 50-60% of transduced cells of different cell lines. We normalized the percentage of transduced cells treated with drugs to percentage of cells not treated with drugs. It was not easy to achieve the same transduction efficiency during the several experiments. To make our results more clear and easy to understand we decided to take the percentage of transduced GFP positive cells not treated with drugs as 100%. An example and the scheme of gating used for Jurkat cells were provided at the Supplementary Figure S2. That means that the percentage of transduced cells not treated with drugs was 55% and the percentage of transduced cells in presence of EME20nM and CQ25uM was 17% when the same virus at the same experiment was used. 55% of GFP positive not-treated with EME/CQ cells was taken as 100% and 17% was normalized and became 31% what was provided at the Figure 2 with bars. The bars at the Figure 2 represent not absolute transduction rate, but relative transduction efficiency. We clarified that in the text of our manuscript, corrected the axis captions and are apologize that this caused the misunderstanding.
Point 4. Now you argue that it is a relative number to untreated cells, then it is normalized to the number of untreated cells. This cannot be true also since the untreated cells do not show any cytotoxicity due to drug treatments, therefore you would expect a huge difference between treated cells and non-treated cells only based on drug cytotoxicity (as you suggested in figure 1, the IC50 for CQ is 12.8 for example for Jurkat cells and 17.64 for EME). The amounts used for Jurkat cells for example (25CQ and 20EME) would kill more than 50% of the cells in the assay and the total number of transfected cells would decrease significantly. Therefore, it seems that the normalization should be done to double drug treated cells not the untreated cells or live cells at the best in the same treatment. Otherwise, this will generate significant results only because of the cytotoxicity alone. This is what I am trying to argue (in the last round of comments also) and still think that the normalization to untreated cells is not the right way to present the data. Some of the synergy calculating programs take into account the cytotoxicity in combination when calculating the effect and can be used to show the data. You can easily visualize this in the pictures in Figure 2B. Imagine you show the number of live cells in the drug treated cells.
Answer to point 4. Thank you, this point is straight forward from the logic when the transduction rate is supposed to be 100%. We are apologize for this misinterpretation. In our study, the transduction rate was not more than 60% that means that both cells in population (transduced and not transduced) are under the cytotoxic pressure of the drugs. This is clearly shown at the Supplementary Figure S2 demonstrating the significant changes in the percentage of cells in population with «normal» FCS and SSC not treated with drugs but transduced with virus (Supplementary Figure S2C) when compared to cells treated with virus and drugs (Supplementary Figure 2E). The percentage of cells with «normal» FCS and SSC decreased twice for the whole population of cells including both transduced and not transduced with virus. Importantly, the percentage of GFP positive cells among the population of studied cells with «normal» FCS and SSC was found to decrease.
By other words, we found that the percentage of GFP positive transduced cells among the whole population of cells significantly decreased while the total number of cells including non-transduced cells also decreased. We suggested that this may be associated with lower viability of transduced cells treated with drugs. All cells in population treated with drugs are dying, but the transduced cells are dying faster, that’s why the percentage of GFP positive cells became lower.
Obviously if the virus did not affect the cell survival, the percentage of transduced cells would not change and would be stable at the base level of 60% in the whole population (or 100% when taken as relative percentage).
That’s why we started to study the synergistic cytotoxic action of EME and CQ in presence and absence of virus and performed apoptosis assays.
The study of antiviral action of drugs was not the main point in our study, that’s why we did not focus on anti-viral drug synergy study. Using several concentrations of drugs we wanted to show the quantitative differences between drugs and its combinations. Which is better presented by bar plots. The main goal of our study was to compare the survival of cells treated with combination of EME and CQ in presence and absence of virus. In this section of study we used SynergyFinder application.
Point 5. It seems that you have selectively shown different treatments in the Figure 2B, D, F, and H than the highest concentrations used in the assay. Could you please also justify why 20EME 10CQ (not the highest) was shown for Jurkat cells but 10EME 5CQ was shown for A549 cells (the lowest concentrations)?
Answer to point 5. We showed several concentrations of drugs to indicate the quantitative effect of the drugs in different combinations on the percentage and MFI. The main goal of this Figure was to demonstrate, that the antiviral potential of drugs can be detected on relatively high concentrations of close to IC50 or higher. Suggesting that, this potential is mostly associated with lower viability of cells treated with combination of drugs and virus.
At the Figure 2 we did not used the combination of 20EME and 10CQ for Jurkat, we showed 5EME/5CQ, 20EME/5CQ and 20EME/25CQ. We used these concentration to demonstrate that non toxic concentrations of 5nM EME and 5uM CQ do not affect transduction efficiency. Only the relatively high dosage of drugs higher than IC50 20EME/25CQ cause antiviral action when Jurkat cells were studied.
For A549 cells the concentration of 5uM CQ was used to show that CQ alone cause antiviral action in low toxic concentration lower than IC50, but the addition of low toxic concentration of EME 10nM do not effect transduction efficiency, only the increased dosage of EME 40nM increase the antiviral effect of CQ, but this concentration is higher than IC50, which possibly points that this action is also associated with cytotoxic action of drugs at this concentrations.
Point 6. When you say (in line number153) “The combination of these drugs causes about three-fold decreases in percentage and MFI of infected cells.” do you mean the expression of the GFP is reduced in infected cells compared to GFP expression in drug-treated cells? In other words, the level of expression is reduced regardless of the absolute number of infected cells. I think you mean the MFI is three folds reduced compared to un-infected cells. If not, then this is the next level of the effect since the expression of GFP is also reduced in the presence of the drugs, which points out to the downstream events in the gene expression when cells are transduced with the GFP coding plasmid LeGO G2.
Answer to the point 6. Thank you for the question. We meant that the combination of these drugs causes about three-fold decreases in percentage of infected cells in the population of cells pre-treated with drugs and the reduction of MFI of the whole population of cells compared to infected cells not pre-treated with drugs. We corrected this sentence in the manuscript to make it more easy to understand. We realize that the drugs may affect the expression and translation of GFP protein and absolutely agree with the reviewer. To exclude this, we verified whether the drugs alone or in combinations affect the fluorescence of GFP positive cells. For that on the base of Jurkat cells we obtained cell line with stable expression of eGFP (GFP encoding lentiviral vector (LegoG2), was used for transduction). In a week after, we treated it with EME and CQ alone and in combinations. We measured the percentage and MFI of cells 72h post treatment and evaluated that the drugs taken in concentrations used in our study do not affect the MFI of cells with stable GFP expression (Supplementary Figure 3). That means that the treatment of cells with drugs followed by virus addition do not affect the GFP expression or translation. That indicates that the reduction of the percentage of GFP positive cells and MFI is mostly associated with decrease of transduction efficiency. The reduction of MFI may be associated with less efficient transgenesis and integration of provirus encoding GFP.
Point 7. Language: Please perform some editing to the text. Examples of which are below:
Line 107-109) The cells were counted after 48h exposition with CQ or EME, and half-maximal inhibitory concentrations (IC50) were determined for both drugs (Figure 1A).
Answer to point 7. We checked our manuscript and edited it including the sentence matched by the reviewer.

This manuscript is a resubmission of an earlier submission. The following is a list of the peer review reports and author responses from that submission.
Round 1
Reviewer 1 Report
Please refer to the attached document.

Author Response
Reviewer 1
Comments and questions
«The authors conducted an impressive study in which they demonstrated that Emetine (EM) can be used in combination with Chloroquine (CQ) and replication-deficient HIV-1 to inhibit cancer cell growth.
This is a well-written article, in my opinion, and the authors presented their research findings clearly and concisely. The findings provided in the manuscript can be significant for the advances in therapeutic approaches to tackle cancer cell proliferation».
Answer. We are very grateful to the reviewer for the warm comments and substantial questions and suggestions.
Point 1. «Why do A549 and H1299 lung cancer cells appear to be unaffected by the addition of RD HIV-1? In some cell lines, it appears as though RD HIV-1 is more adept at utilizing autophagy pathways than other viruses. I would recommend that the authors briefly explain the differences seen in the effect of RD HIV-1 on the proliferation of the four different cancer cell lines addressed in the text».
Answer. The main point of our study is that lentiviral particles based on HIV-1 do not cause any cytotoxic action on cells when added alone. This is true for all four cell lines used in our study. We found that RD HIV-1 significantly induce/increase the cytotoxic action of Emetine and Chloroquine when added all together. But, we were also surprised that the synergistic effect of drugs and RD HIV-1 was significantly less pronounced when added to lung carcinoma cells A549 or H1299. Truly, we are not sure that we have enough experimental data to suggest the mechanisms responsible for less sensitivity of A549 and H1299 cells compared to Jurkat and Hek293. Absolutely clear that it depends on the differences of these cell lines. Possibly, the most pronounced action detected on Jurkat cells is associated with the fact that this is T-leukemia/lymphoma cells which is initially more close to the target cells of HIV-1. Possibly there are more mechanisms involved in cell survival which are also utilized by viral particles based on HIV-1. Contrary, the HEK293 cells obtained from embryonic kidney were found to be highly sensitive to Emetine/Chloroquine and RD HIV-1 combination. But these cells are widely used in different cancer studies and furthermore were demonstrated to have similarities to neuronal cells and are used in neuronal studies. Interestingly, earlier it was shown that neuronal cells are highly sensitive to HIV-1 viral proteins and components of medium from the cells used for HIV-1 production https://doi.org/10.1073/pnas.0304859101. Possibly, the similar mechanisms may be activated when HEK293 cells were used in our experiments. Interestingly, we did not determine any impact of RD-HIV-1 Emetine and Chloroquine on apoptosis activation in H1299 cells. We suggested that this may be associated with the p53-deficiency of these cells. But truly, these speculations are not based on the strong experimental data and more cell lines should be examined to suggest the mechanisms. That’s why we would not like to add such kind of explanations into the manuscript. But we noted that H1299 cells are P53 deficient in section describing the apoptosis experiments.
Point 2. «The authors treat cancer cells with EM and CQ first, followed by the addition of RD HIV-1. Because both EM and CQ have been shown to inhibit cancer cell growth, the addition of RD HIV-1 following EM+CQ treatment appears to have an additional effect on decreasing cell growth. How about first infecting the cells with RD HIV-1 and then treating them with EM+CQ? Is it likely that the synergistic impact of RD HIV-1, EM, and CQ on cancer cell growth will persist? Please provide a few sentences in the text addressing this».
Answer. Yes, before starting the experiments we tried different orders of drugs and virus adding. Generally, the action was mostly similar, but the scheme when the virus first and followed by drugs addition demonstrated less pronounced action. That’s why, we decided that the oncolytic action of virus is mostly associated with first stages of infection. When we added the drugs to the cells 24 hours post-transduction we did not found any additional cytotoxic activity of RD HIV-1. That possibly means that this action is implemented on the first stages of virus life cycle prior to integration into the host cell genome. As recommended by the reviewer, we added this sentences into the text.
Point 3. «What I find interesting is that RD HIV-1 diagonally shifts the viability of the cells as seen in the matrices provided in Figures 3 and 4. Thus, RD HIV-1 appears to have a similar effect to low doses of EM and CQ, but with a similar EM/CQ ratio of higher doses of EM and CQ (at which the cell viability is very low). Thus, RD HIV-1 might theoretically be used in place of EM and CQ, provided that its infection effectiveness can be controlled. It would be great if the authors could give additional insight on the possibility of using RD HIV-1 alone to reduce cancer cell growth and the complications that might arise».
Answer. Thank you very much for this comment. This observation is very valuable and deserve the further investigation. The HIV-1 inhibitors may be used to modulate the RD HIV-1 trunsduction efficiency. Generally we suppose that some types of cancers may be attenuated by the combination of HIV-1 inhibitors and RD HIV-1. Possibly, some types of cancers are less frequent in HIV-1 patients treated with HIV-1 inhibitors. This is really very interesting, but we would not like to touch that prior the accurate study, which should be conducted to evaluate these patterns.
Reviewer 2 Report
The manuscript “Emetine synergistically with Chloroquine induces oncolytic potential of HIV-1 based lentiviral particles.” shows the synergistic effect of the Chloroquine and Emetine to kill various cell lines which have been transformed to become immortalized. Also, the antiviral activity of the combination is discussed against HIV1 as well as the induced cell death by HIV1 in the presence of the two drugs. The manuscript has interesting data but have some major points to take in before considered for publication.
There are minor and major points which are mentioned in the order of appearance in the manuscript:
The use of EM for Emetine is a bit confusing since it is generally known for electron microscopy. Could you consider using another abbreviation for that?
Some of the antiviral evidence in the case of Chloroquine for example for SARS-CoV is very subjective and should be mentioned in here since it was debated a lot and is not under consideration anymore.
The Chloroquine has mostly been proposed as an adjuvant therapy in cancer treatment but not being considered as the main treatment. It also has apoptosis and necroptosis and immunomodulatory or anti-inflammatory properties which are unrelated to anti-autophagy effect of the Chloroquine. Additionally, the clinical efficacy and safety has not been established for chloroquine yet, that should be acknowledged.
It has been shown that “Chloroquine inhibits cell growth in human A549 lung cancer cells by blocking autophagy and inducing mitochondrial‑mediated apoptosis” by Likun Liu et al. in 2018.
The line cells are normally transduced with oncogenes and protooncogenes from viral sources. Therefore, the comparison to a real-life condition is questionable.
The figure 1 panel B is wrongly marked as Emetine but should be Chloroquine. The scale of 0.01 to 1 and to 100 does not seem good since the changes in the concentration is more than 100 times. Please change it to shorter range or increase the X axes values.
The effect of the virus on cell viability has been tested in the presence of the drugs and have been compared to mock infected cells pre-treated with the drugs but the effect of the virus alone is missing and is required to make a proper comparison.
The virus used in the experiment needs to be purified not to carry any additional element from the preparation such as miRNA or any cytokine since the effect might be due to the other factors than the virus itself and the drugs used.
The whole claim relies on the comparison between the apoptotic portion of infected and non-infected cells, and this relies on the expression of the GFP in infected cells to identify the infection. Now the question is the efficacy of expression among infected cells. How can you identify the efficacy of translation of the GFP gene among infected cells?
The percentage of the apoptotic cells in two fractions, infected and non-infected, also contains cell proliferating. How the proliferation differs between these fractions to influence the final proportions calculated in each fraction?
You have reported the cell death as well as the cell viability in synergy plots. How is the data defined in the analysis of the synergism? Have you defined the antiviral effect using the viability and the cell death effect using the 100 minus viability? Please specify.
The overall verdict is that the applicability of the proposed method in patients might be a long shot since the efficacy of the anti-cancer treatment relies on multiple rounds of treatments in certain intervals and using the RD-HIV1 might elicit neutralizing antibodies and reduce the efficacy of the treatment (please see doi:10.1006/mthe.2000.0116). Additionally, the specificity of the virus to cancer cells is under question and there is no guaranty that the tumor or cancer cells can be easily targeted by RD-HIV1.
Author Response
Reviewer 2
Comments and Suggestions for Authors
The manuscript “Emetine synergistically with Chloroquine induces oncolytic potential of HIV-1 based lentiviral particles.” shows the synergistic effect of the Chloroquine and Emetine to kill various cell lines which have been transformed to become immortalized. Also, the antiviral activity of the combination is discussed against HIV1 as well as the induced cell death by HIV1 in the presence of the two drugs. The manuscript has interesting data but have some major points to take in before considered for publication.
Answer. We are very grateful to the referee for a thorough and critical analysis of our manuscript. We have made all the necessary corrections in the text of the manuscript. We also provide answers to questions posed by the reviewer.
Point 1. «The use of EM for Emetine is a bit confusing since it is generally known for electron microscopy. Could you consider using another abbreviation for that?»
Answer. As recommended by the reviewer, we have changed the abbreviation for emetine. EM has been replaced by EME in text and in Figures.
Point 2. «Some of the antiviral evidence in the case of Chloroquine for example for SARS-CoV is very subjective and should be mentioned in here since it was debated a lot and is not under consideration anymore.»
Answer. We agree with the reviewer. In introduction section we mentioned different aspects of anti-viral action of Chloroquine in context of different viruses including SARS-Cov-2 and provided the links to the sources. Furthermore in introduction section of our manuscript we noted that the antiviral potential of Chloroquine in case of SARS-Cov-2 is contradictive.
Point 3. «The Chloroquine has mostly been proposed as an adjuvant therapy in cancer treatment but not being considered as the main treatment. It also has apoptosis and necroptosis and immunomodulatory or anti-inflammatory properties which are unrelated to anti-autophagy effect of the Chloroquine. Additionally, the clinical efficacy and safety has not been established for chloroquine yet, that should be acknowledged.»
Answer. We agree with the reviewer. In provided study we do not touch the aspects of clinical efficacy of Chloroquine in anti-cancer therapy and do not raise the issue of using chloroquine as a main treatment. The main point of our study – is that the use of Emetine in combination with chloroquine may induce apoptosis of cancer cells treated with medium containing replication deficient HIV-1 based lentiviral particles. Furthermore we do not discuss the aspects of its safety. Importantly chloroquine was used as an antiprotozoal drug for many years and several clinical trials of its potential application in anti-cancer therapy are undergoing. (Liu Ting et al. Front. Pharmacol, 2020). Thus the features of its anti-cancer action when used in combination with other agents deserve the interest, that’s why we decided to describe the phenomenon we accidentally discovered.
Point 4. «It has been shown that “Chloroquine inhibits cell growth in human A549 lung cancer cells by blocking autophagy and inducing mitochondrialmediated apoptosis” by Likun Liu et al. in 2018.»
Answer. We are grateful to the reviewer for his recommendations. We have included the link to this source into the text of our manuscript in section, where the mechanisms of anti-cancer action of chloroquine are discussed.
Point 5. «The line cells are normally transduced with oncogenes and protooncogenes from viral sources. Therefore, the comparison to a real-life condition is questionable.»
Answer. We are agree. Importantly, cell lines and human cells in real organism bears a lot of endogenous viral sequences that can be derivated from retroviruses. Therefore, this is one of the circumstances which restrict the experimental data interpretation. The cytotoxic activity of lentiviral vector stocks may depend on many factors, including the medium used for viral particle generation, condition of cells used for viral particle production, multiplicity of infection, insertion mutagenesis and many others. In our study, we represent the phenomenon, which may be of interest for the researchers studying the different aspects of lentiviral vectors application. We are agree that the translation of the effects described in this manuscript may differ from real-life condition and think this deserve a detailed and separate study.
Point 6. «The figure 1 panel B is wrongly marked as Emetine but should be Chloroquine. The scale of 0.01 to 1 and to 100 does not seem good since the changes in the concentration is more than 100 times. Please change it to shorter range or increase the X axes values.»
Answer. Thank you, we corrected the axis labels. IC50s were calculated by nonlinear regression with variable slope (four parameters) and robust fitting using The GraphPad Prism software v.9.3.1. The graphs represented on Figure 1 given to represent the slope of the curves and the range of concentrations used for IC50 calculation. We tried to change the scale, andit changed out that the provided scaling was found to be the most suitable based on the logic of IC50 calculation algorithm.
Point 7. «The effect of the virus on cell viability has been tested in the presence of the drugs and have been compared to mock infected cells pre-treated with the drugs but the effect of the virus alone is missing and is required to make a proper comparison.»
Answer. We used the lentiviral vectors based on the the third generation self-inactivating viral vectors LeGO. The lentiviral vectors were generated as described in protocol (http://www.lentigo-vectors.de/protocols.htm). These vectors were previously used in different studies on various cell lines and were not shown to effect cell survival when used even in relatively high multiplicity of infection protocols. In our study we used the viral vectors to achieve not more than 50% of transduced cells, what is up to MOI1 in case of HEK293, Jurkat, H1299 and A549 (used in our study). Furthermore, in our study, we tested the viability of virus containing medium when added alone in absence of inhibitor (Figure 3 (A) and Figure 4 (A) – dose response matrix). In experiment representing the effect of drugs on cell survival we used as a control the sample of cells non –treated with drugs and non-treated with virus-containing medium (left «control» panel Figure 3(A)). The effect of drugs in absence of virus was compared to the effect of drugs in presence of virus (left «virus» panel Figure 3(A). In this experiment we used as a control the sample not treated with drugs, but treated with virus containing medium. The dose response matrix used in Synergy finder algorithm represent both controls (no drugs/no virus and no drugs/plus virus) expressed as 100% (not treated with drugs). But initially we obtained the data for both controls and there was no significant differences found between them.
Point 8. «The virus used in the experiment needs to be purified not to carry any additional element from the preparation such as miRNA or any cytokine since the effect might be due to the other factors than the virus itself and the drugs used.»
Answer. Thank you very much, we are absolutely agree. We decided to change the text of the manusctipt to refrain from using the term «virus» in favor of «medium containing lentiviral particles» or «viral stocks», corrected the discussion section of our manuscript and pointed the aspects noted by the reviewer. Contrary, different methods of virus purification also has disadvantages which may complicate the interpretation of results. We deceided to verify the obtained results using the system of lentiviral particles pseudotyped with two different envelope proteins. The first on is VSVG and the second one is McERV. McERV envelope protein use proteolipid plasmolipin for cell entry. The Jurkat cells lack this proteolipid on cell surface and could not be transduced by lentiviral particles pseudotyped with McERV envelope protein. The HEK293 cells may be efficiently transduced because this cells carry plasmolipin exponated on cell surface. This means that treatment of Jurkat cells with lentiviral particles pseudotyped with VSVG in presence of drugs should effect cell survival, but the lentiviral particles pseudotyped with McERV envelope protein should not. Contrary HEK293 cell survival should be effected by both variants of lentiviral particles when added in combination with drugs. Based on the results which we have already obtained – most likely, the viral particles affect the cell survival survival, but not the additional elements such as cytokines, miRNA or exosomes content. But this experiments are still undergoing, so we cannot exclude this. We are very grateful to the reviewer for this point.
Point 9. «The whole claim relies on the comparison between the apoptotic portion of infected and non-infected cells, and this relies on the expression of the GFP in infected cells to identify the infection. Now the question is the efficacy of expression among infected cells. How can you identify the efficacy of translation of the GFP gene among infected cells?»
Answer. The question is absolutely fair, indeed, the effect of drugs on the level of gene expression in cells, including the expression and translation of GFP, cannot be ruled out. The main statement of the work is related to the fact that a decrease in the number of transduced cells and a decrease in the intensity of their fluorescence is associated with a decrease in their survival when combinations of the drugs we study are used. In this experiment, the presence of fluorescence, regardless of its entencity, was chosen as a marker, which made it possible to separate transduced cells from non-transduced ones (not expressing the green protein at all). Next, we performed the detection of apoptotic cells in both populations. This experimental design does not imply a comparison of flourescence intensity. Here the presence of GFP is a discrete value, necessary simply to separate the transduced cells. But in general, the intensity of fluorescence in apoptotic cells obviously should be and always is lower when compared to healthy cells, which are less susceptible to toxic effects, and the reviewer's question is absolutely fair and significant.
Point 10. «The percentage of the apoptotic cells in two fractions, infected and non-infected, also contains cell proliferating. How the proliferation differs between these fractions to influence the final proportions calculated in each fraction?»
Answer. Here, we compared the percentage of apoptotic cells in the transduced and non-transduced fractions. Despite the fact that the ratio of actively proliferating cells may also differ, we were interested in the final contribution to the decrease in survival, which is expressed precisely in the activation of apoptosis. Probably, the apoptosis is a the result of cell cycle arrest associated with the action of the virus and the drug, as well as many other mechanisms. We are grateful to the reviewer and will certainly take this remark into account when designing the experiment of our other works.
Point 11. «You have reported the cell death as well as the cell viability in synergy plots. How is the data defined in the analysis of the synergism? Have you defined the antiviral effect using the viability and the cell death effect using the 100 minus viability? Please specify»
Answer. To obtain synergy plots, we used the numbers of cells, determined by discrimination of viable cells by trypan blue staining in neubauer chamber. The number of treated and non-treated cells was analyzed and the dose response matrix was obtained to calculate synergistic action. For that we used on line software (https://synergyfinder.fimm.fi). The antiviral effect of Emetine and chloroquine was described in the first part of the manuscript. The efficiency of (infection) trunsduction in presence of drugs was detected by FACS, by the analysis of percentage and MFI of transduced cells. This system was previously published and is widely used to study the antiviral drugs acton https://www.ncbi.nlm.nih.gov/pmc/articles/PMC3560153/. The lentiviral particles in concentrations used in our study do not cause any toxic action on cells, and we did not used the viability assays to determine the infection efficiency. The main point of our study is that the antiviral action of drugs is mostly associated with the reduced cell survival, which is synergistically activated by combination of virus and both drugs.
Point 12. «The overall verdict is that the applicability of the proposed method in patients might be a long shot since the efficacy of the anti-cancer treatment relies on multiple rounds of treatments in certain intervals and using the RD-HIV1 might elicit neutralizing antibodies and reduce the efficacy of the treatment (please see doi:10.1006/mthe.2000.0116). Additionally, the specificity of the virus to cancer cells is under question and there is no guaranty that the tumor or cancer cells can be easily targeted by RD-HIV1.»
Answer. In this work, we did not touch upon the issues related to the possibility of using this approach in the treatment of malignant diseases and, moreover, the technical aspects of the use of lentiviral vectors in therapy. It should be noted that there are different approaches to increase the tropism of lentiviral vectors to different cancer cells of different origins and third-generation, self-inactivating lentiviral vectors have been used in multiple clinical trials to introduce genes into hematopoietic stem cells to correct primary immunodeficiencies and hemoglobinopathies. But we are agree that the questions of the lentiviral vectors stability, efficiency and safety are very discussive. Here we used the third-generation LeGO vectors, which were successfully used by other groups both in vitro and in vivo and performed well in this type studies.